# Item-Level Psychometric Analysis of the Psychosocial Processes at Work Scale (PROPSIT) in Workers

**DOI:** 10.3390/ijerph19137972

**Published:** 2022-06-29

**Authors:** César Merino-Soto, Arturo Juárez-García, Guillermo Salinas-Escudero, Filiberto Toledano-Toledano

**Affiliations:** 1Instituto de Investigación de Psicología, Universidad de San Martín de Porres, Lima 34, Peru; sikayax@yahoo.com.ar; 2Centro de Investigación Transdisciplinar en Psicología, Universidad Autónoma del Estado de Morelos, Pico de Orizaba 1, Los Volcanes, Cuernavaca 62350, Mexico; arturojuarezg@hotmail.com; 3Centro de Estudios Económicos y Sociales en Salud, Hospital Infantil de Mexico Federico Gómez, National Institute of Health, Márquez 162, Doctores, Cuauhtémoc, Mexico City 06720, Mexico; guillermosalinas@yahoo.com; 4Unidad de Investigación en Medicina Basada en Evidencias, Hospital Infantil de México Federico Gómez, National Institute of Health, Márquez 162, Doctores, Cuauhtémoc, Mexico City 06720, Mexico; 5Unidad de Investigación Sociomédica, Instituto Nacional de Rehabilitación Luis Guillermo Ibarra Ibarra, Calzada México-Xochimilco 289, Arenal de Guadalupe, Tlalpan, Mexico City 14389, Mexico

**Keywords:** stress, psychosocial risk factor, assessment, content validity

## Abstract

The structural attributes and correlates of items have an effect on their composite scores and exploring them strengthens the content validity of a measure adapted to another context. The objective of this study was to evaluate the item properties of a measure of psychosocial work factors (PWFs). Data were collected through a web platform from 188 Peruvian working adults (men = 101, 50.5%) holding various professions and jobs. The instrument was the Psychosocial Processes at Work Scale (PROPSIT), adapted for the Peruvian context. The distributional characteristics, the efficiency of its response options and its correlates with engagement, occupational self-efficacy, general stress and psychological distress (explored with a coefficient of maximum information and another of monotonic association) were analyzed. It was found that the items were asymmetrically distributed, without statistical normality and with a response tendency at low (for psychosocial risk factors (PSRFs)) and medium (favorable psychosocial resources) levels. The number of efficient response options was lower (approximately five options) than the original structure (seven options). The monotonic associations with gender and age were essentially zero and theoretically converged with the external constructs, except for some items related to job demands. The contributions of the results to the content validity of the PROPSIT and the orientation of working hypotheses about PROPSIT item constructs and measures of work effects are discussed.

## 1. Introduction

In the creation and/or adaptation phase of the measures, detailed univariate item information is considered good practice for reporting quantitative results and serves to better understand the quantitative functioning of the scales or tests, and for the characterization of the measured construct [1,2]. By extension of the above and more specifically, the properties of univariate items can be extremely useful in revealing the response tendency of individuals being evaluated, the location of items in the observed score range and attribute variability in the specific behaviors captured by the items. For example, in a measure that contains items with different strengths of association with the criterion of interest, its predictive ability with the criterion may also vary. Regarding the latent attribute in these items, the factor or observed score will consequently show a decrease in its predictive potential with these items [3].

Item validity research usually focuses on the item–construct relationship within the instrument itself, commonly referred to formally as the internal structure [4]. In a multidimensional instrument, the internal structure assesses and describes the item’s association with its own dimension and other dimensions. This source of validity evidence is aligned with two properties of the internal structure of a construct: Its convergent validity and divergent validity [5,6], that is, the strength of the variance in items within their construct versus the variance in other factors that do not attempt to measure it. An additional step can be found by assessing the scalability of each item, which is usually represented in estimates based on the Rasch model [7], where items and latent attributes are associated on the same scale.

The development of these aspects helps to characterize the validity of items to represent their construct within the conceptual content of the instrument, but item validity can be extended to include their association with external criteria and other structural features of the items, such as the functioning of their response options. For example, if items created to assess workplace bullying theoretically should be associated with symptoms of depression (e.g., [8,9]), then this covariability should be observed not only in the total scale but also in estimates of statistical dependence between the items and empirical criteria of depression, specifically identifying the size of these associations, as well as their positive or negative direction and their statistical significance. This external item–criterion property has been referred to as item validity [3,10], yet it seems to be least addressed in the construction or adaptation of instruments in contemporary research, specifically in research on psychosocial work factors (PWFs). Indeed, this point is corroborated in the reduced treatment of this topic in assessment and psychometrics books (e.g., [11], whole book; [12] chapter three; [10], chapter six; [3], chapter six) or in current good practice recommendations for measure validation (e.g., [13]). The network of associations of items with other constructs outside their measurement model gives shape and meaning to what is known as a nomological network [14], in which relationships between constructs are hypothesized, analyzed, and verified by constructing expected relationship hypotheses. At the item level, this means configuring convergent (*r* ≠ 0) or divergent (*r* ≅ 0) relationship hypotheses about the items with other constructs. The strength of item validity in these aspects has relevance for interpreting the score of the instrument that the items are part of [3] because it highlights the theoretical and applied value of the content of the items.

When addressing the issue of association between variables, one aspect that is introduced is the differentiation between linear and nonlinear dependence. In research on bivariate dependence between variables during the construction and adaptation of measures in the social sciences, explorations of linear relationships predominate [15]; however, it is reasonable that other types of association exist. In the detection of bivariate associations, linear, monotonic, and nonmonotonic associations may exist [16], and psychological research has not ignored explorations of different associations beyond the single linear dependence between variables. In the psychological literature on developmental constructs and in other applied contexts, nonlinear relationships have emerged as a covariance model that validly represents the association between variables [17], for example, in variables such as the development of visuomotor skills in a sensitive age range of change (e.g., [18]), in role, status and personality change in adults (e.g., [19]) and in the expression of depressive symptomatology (e.g., [20]). In the work context, the relationships between stress with coping skills (e.g., [21]) and work self-efficacy [22] and on-job performance and job insecurity and negative affectivity [23] have also been explored. Clearly, however, the identification of monotonic and nonmonotonic nonlinear associations seems to be rare.

In exploratory contexts where the evaluation of statistical dependence between variables is performed, it is especially useful to detect the type of bivariate functional relationship that exists [24,25], without necessarily starting from a predefined hypothesis about some type of association, as usually occurs when starting with the search for linear relationships [16,24,26,27]. This point implies that the search for bivariate relationships should start by detecting some kind of existing association and serve as a condition for performing a second step where more specific or more complex relationships are tested (e.g., multivariate analysis), based on either a linear or nonlinear model. In this sense, the incorporation of modern methods to detect a wide range of bivariate associations also expands the knowledge about the construct measured, as well as the performance of the items when adapting psychosocial measures of the work environment.

Particularly in the item analysis of measures of PWFs, there is evidence that the symptomatology of psychological effects at the individual level can serve as a marker for the effect of psychosocial factors such as effort/reward imbalance, limited perceived support, high job demands, emotional demands or mobbing. Such markers have included anxiety symptoms [28,29,30], depressive symptoms [9,29,30,31,32,33] and general mental health status [29,34,35,36]. On the other hand, individual variability derived from constructs such as job self-efficacy has been considered a moderating factor in the relationship between factors psychosocial of work and positive and negative affect [37], and job engagement has been an indicator associated with job demands and resources [22,38]. Taken together, these constructs can be good markers of item validity when considered in the stage of developing and adapting a measure.

Item analysis would not be complete without examining the structural characteristics of the items, which are exactly centered on their response options. In this field, there are heuristics regarding the optimal characteristics to ensure the quality of the response options (e.g., [39,40]), which together describe their functionality and influence the scoring performance and the validity of the internal structure of the instrument [41]. For example, these heuristics correspond to the “attractiveness” of each option, the regular distributional form of responses, the functional number of options, the monotonicity in relation to the scale score and other options, which can be considered as conditioning the effectiveness of the chosen ordinal system [40,41,42]. Since the performance of a scale should converge with the optimal performance of its items, to make research on the content of the items complete, the validity of the items with external criteria and the performance of their response options should be integrated into the analysis of the items.

In the present study, the methodological aspects above (i.e., the exploration of nonlinear associations, the nomological network and the quality of the ordinal system) were applied to adapt a new measure of work factors and processes, the scale of the Psychosocial Processes at Work Scale (PROPSIT; [43,44]), to the Peruvian context. The PROPSIT was created to represent the main factors of the work context (e.g., psychological demands, control, rewards) extralabor factors (e.g., family stressors, traffic transfer), mediating psychological effects (e.g., burnout, engagement) and relevant mental health variables (e.g., somatic symptoms and alterations including depression, anxiety, posttraumatic stress) in Mexican workers. From the primary prevention approach, the most important subscales of the PROPSIT are PWFs, which recognize the existence of negative factors (psychosocial risk factors [PSRFs]) and positive factors (favorable psychosocial resources). The theoretical framework for this measure was the demands–resources model [45,46,47]. Additionally, the measure was constructed with a culturally relevant view to its Mexican context of origin, but it is potentially generalizable to other Latin American contexts due to the theoretical analysis of factors with etic value (i.e., factors consistently identified in several international studies; [48]) and that are consistently chosen as core attributes to assess PSRFs.

Notably, the classical approach of determining the psychometric properties (e.g., factor analysis and reliability estimation) of a measure such as the PROPSIT, although methodologically related, can be independent of the detailed psychometric exploration of the behavior of its items in terms of the identification of linear or nonlinear relationships, the identification of their nomological network and their overall item quality. In a strict sense, the results can be part of research on content validity from a quantitative perspective to assess the psychometric properties at the item level.

Given the above, the aim of the present study was to identify the structural properties of the items of the PROPSIT work factors scale as well as their convergent and divergent associations with constructs of psychosocial effects on Peruvian workers. These properties included the functionality of the response options and their expected ordering as well as their distribution function. Furthermore, through an analysis of the associations of items with variables external to the measurement model of the instrument, the validity of each item was evaluated with measures of psychosocial effects and individual processes theoretically relevant to the PROPSIT measurement model (nomological network). For example, the workload dimension can be expected to be incrementally associated with job dissatisfaction or stress [35]; therefore, the items of this measure should maintain positive monotonic associations with external measures of job dissatisfaction or stress.

## 2. Materials and Methods

### 2.1. Participants

The sample of participants was obtained nonprobabilistically; the eligibility criteria were as follows: Having reached the age of majority, being a worker with a formal contract and having current work activity (without vacations, sick leave, etc.). Table 1 shows the characteristics of the workers. The sample consisted of 188 workers holding various occupations and careers; 69.7% were workers with university studies, 20.2% were workers with technical studies and 9.6% were workers with basic studies. According to the International Standard Classification of Occupations (ISCO-08; [49]), the participants predominantly (>50%) held mid-level technical and professional occupations (see Table 1).

### 2.2. Instruments

#### 2.2.1. Psychosocial Processes at Work Scale (PROPSIT)

The PROPSIT [43] is a measure of factors, effects and intervening processes of work for assessing psychosocial work conditions and the effects on the well-being of the worker. It contains four general sections: PWFs, the psychosocial effects of work, the health-disease process and other extrawork psychosocial factors. The present study validated the psychosocial factors at work section, consisting of two major parts: Favorable resources, with 19 items distributed across the dimensions of rewards and work development (7 items), work control (5 items), resources to do the job (2 items), work social support climate (3 items) and value congruence (2 items); and psychosocial risks, with 22 items distributed across the dimensions of workload and pace demands (2 items), demands of high responsibility and dangerousness (3 items), demands of working hours, shifts or schedules (3 items), cognitive or attentional demands (3 items), emotional demands due to dealing with people (3 items), demands of physical effort and the physical environment (3 items), psychological harassment at work (2 items) and stressful leadership (2 items). In a previous construction study [43], the *α* and reliabilities ranged between 0.73 and 0.85 and between 0.70 and 0.87, respectively, with little distance between them.

#### 2.2.2. Occupational Self-Efficacy Scale (OSES)

The OSES [50] is a six-item instrument that assesses self-efficacy in the global work context through the perception of sufficiency and confidence in the ability to complete tasks at work. Satisfactory cross-cultural validity was found in the original study [50], and the high score reliability and theoretical relationship with work criteria and other constructs were corroborated in a Peruvian context [22]. In the present study, the reliability was satisfactory, *ω* = 0.95 (95% confidence interval [CI] = 0.92, 0.96).

#### 2.2.3. Utrech Engagement Scale (UWES-3)

The UWES-3 [51] assesses work enthusiasm, with three items on vigor, dedication and absorption chosen on a rational and empirical basis by the authors and derived from the UWES-9. The response scale is seven points, ranging from 0 (never) to 6 (always). In two Peruvian studies, validity results confirmed its construct validity with respect to the internal structure, reliability and relationship with other variables (Calderón-de la Cruz et al., in preparation; Merino-Soto et al., in preparation). In the present study, the internal structure was adequate, *ω* = 0.94 (95% CI = 0.91, 0.96).

#### 2.2.4. Single Stress Item (SUI)

The SUI [52] is a single-item measure created to assess stress symptoms in a global frame of reference, ordinally scaled with five options (from not at all to very much). In general, it is an efficient measure for approximating the experience of stress, and in particular, it has been used in studies on work stress [52,53,54,55,56].

#### 2.2.5. Patient Health Questionnaire-4 (PHQ-4)

The PHQ-4 [57] is a brief screening measure of emotional and cognitive symptoms of depression (two items) and generalized anxiety (two items), and it has been internationally accepted as a total screening measure of efficient psychological distress. It is scaled with a five-point scale (from not at all to almost every day) regarding the participant’s experience within the last two weeks. In studies with Peruvian participants [58,59], its correlations maintain theoretical consistency with other constructs. In the present data, the fit to two correlated dimensions (i.e., anxiety and depression) was satisfactory, WLSMV-*χ*^2^ = 0.067 (df = 1, *p* = 0.79) CFI = 1.00, SRMR = 0.00, WRMR = 0.061, but the interfactor correlation was very high (*r* = 0.94, 95% CI = 0.88, 0.99). Therefore, a single score (WLSMV-*χ*^2^ = 3.01, df = 2, *p* = 0.22, CFI = 1.00, SRMR = 0.02, WRMR = 0.409; factor loading: 0.89, 0.90, 0.95 and 0.93), interpreted as psychological distress [57], was used. The internal consistency of the score was *α* = 0.90 (95% CI = 0.85, 0.93).

#### 2.2.6. Ethical Considerations

This study is a part of a research project (HIM/2015/017/SSA.1207; “Effects of mindfulness training on psychological distress and quality of life of the family caregiver”) that was approved by the Research, Ethics and Biosafety Commissions of the Hospital Infantil de México Federico Gómez, National Institute of Health, in Mexico City. While conducting this study, the ethical rules and considerations for research with humans currently enforced in Mexico [60] and those outlined by the APA [61] were followed. All participants were informed of the objectives and scope of the research and their rights in accordance with the Declaration of Helsinki [62]. Participants who agreed to participate in the study signed an informed consent form. Participation in this study was voluntary and did not involve payment and was conditioned upon completion of the informed consent form.

### 2.3. Procedure

#### 2.3.1. Data Collection

The call for participants was made through the social networks of the principal investigator and two Peruvian collaborators, undergraduates that held degrees in psychology; the social networks used were WhatsApp and Facebook. Eligibility for the selection of contacts was based on age (a minimum of 18 years) and work activity (working at the time of data collection). The first contact with the participants was to explain briefly in writing what the study consisted of and the content of the survey; then, the participants received the survey link through the same means of communication. Participants were also invited to forward the link to their own contacts. The material was arranged in a single sequence for all: The participation consent form (with information on the objective of the study, the voluntariness of participation, the anonymity of responses, freedom to stop responding at any time, confidential treatment of data and absence of participant traceability). The selection of the participants after data collection was based on (a) contractual employment with a Peruvian employer in the last job, (b) a minimum work time of six months and (c) voluntary acceptance of participation through the means of the consent form.

#### 2.3.2. Analysis

Univariate descriptive, structural (use of response categories, response similarity, scaling distribution) and associative (relationship with gender, age, occupational self-efficacy, engagement and psychological distress) measures were obtained. As a frame of reference, the structural characteristics had the aspects necessary to establish the structural quality of the items [40,41,42].

#### 2.3.3. Descriptive Properties

Univariate measures of TE response trend or intensity, skewness, kurtosis and statistical normality were obtained and examined with the Anderson–Darling test at the nominal 0.15 level [63]. This test can be considered efficient for ordinal variables [63]. The *MVN* statistical package from software R [64] was used.

#### 2.3.4. Efficiency of Use of Response Categories

First, the absolute frequency of responses in each category was recorded, and categories with fewer than 10 observations were reported as problematic or inefficient [40]. To guarantee the precision and stability of the psychometric estimates in each response option, this number is the minimum recommended size [41,42]. Second, the number of effective response options was identified (*actual equivalent number of options* [AENO]; [65]). Similarly known as the *number of equivalent hypothetical alternatives* [66], it is a quantitative indicator of the number of functional choices that actually occur in the sample. It is formulated in the conceptualization of entropy as follows:
$$\mathrm{AENO}\;=\;2^{-\sum_{i=1}^{k}p_{i}\log_{2}(p_{i})}$$

In the formula, *k* is the number of options, *p* is the proportion of responses in option *k* and log is the base 2 logarithm. AENO represents the maximum amount of information, defined as the number of different response options. Options for which the frequencies were zero were treated as incidental zeros [40], and they were replaced with 0.05.

#### 2.3.5. Scaling Distribution

The items were treated as ordinal categorical variables to examine their distributional properties. First, the AJUS approach [67] was applied to identify the pattern of responses using four classificatory categories: A (i.e., unimodal), J (i.e., unimodal with elevation in either tail), U (i.e., bimodal with elevation in both tails) and S (i.e., multimodal).

Second, the degree of response similarity within the same item was estimated by a measure of consensus for ordinal variables, the Tastle–Wierman (*TW*) coefficient (varies between 0 and 1.0; [68]); values close to 1.0 indicate greater consensus or response similarity. The *agrmt* statistical package from software R [69] was used for these analyses.

#### 2.3.6. Association with Sociodemographic Variables

To detect possible covariation with respect to gender, we used the biserial rank correlation (rbr; [70]), implemented with the *rcompanion* (*wilcoxonRG* function; [71]), with a 95% CI. With respect to age, to detect some form of monotonic dependence, the Spearman correlation coefficient was used.

#### 2.3.7. Association for Construct Validity

To identify convergent and divergent item relationships, the OSES (i.e., occupational self-efficacy), enthusiasm (i.e., UWES-3), stress (i.e., SUI) and psychological distress or mental health symptom (i.e., PHQ-4) measures were used as criteria. To detect dependencies in a wide range of covariation (e.g., linear monotonic, nonlinear monotonic and nonmonotonic), the *maximum information coefficient* (MIC; [72]), which comes from the maximal information-based nonparametric exploration (MINE, [25]) approach, was used. Compared to other associative measures (i.e., correlation distance), this nonparametric measure shows adequate power to detect a wide range of dependence structures [26,27]. The MIC provides a value between 0.0 (complete statistical independence) and 1.0 (complete dependence), indicating the strength of the relationship for some type of bivariate function relationship (not necessarily linear). Because it can be compared to the standardized covariance percentage (*r*^2^) and measures of mutual information tend to be inefficient for detecting linear relationships [17,24], it was used in conjunction with the Spearman monotonic correlation to detect monotonic covariation. We used the R program *testforDEP* [73].

## 3. Results

### 3.1. Descriptive Information for Items

#### 3.1.1. Psychosocial Risk Factors (PSRFs)

In general, the intensity of responses (mean response) varied consistently in the range of the options “never” and “regularly (a few times a month)”, except in the cognitive/attentional, workload and work rhythm and negative feedback areas, where the trend was higher. On the other hand, the difference in mean response within each content area was noticeable in the workload and work rhythm and activity schedule (shift and schedule) areas, suggesting conceptually heterogeneous contents. Except for these contents, in general, the response trend was similar in the PSRF content areas themselves. Variability was consistent within each content area and generally varied moderately (SD_max_/SD_min_ = 2.38).

The trend of skewness and kurtosis was consistent in each content area, except in some, where there was observable variability in direction (e.g., activity schedule). Statistical normality (AD statistic) was not accepted in any item and was of similar magnitude in each content area; the exception was the shift and schedule area, where the degree of nonnormality varied considerably. On the other hand, the degree of response agreement was moderately high (*Tw* > 0.50), indicating the predominance of unimodal and regular distributions.

#### 3.1.2. Psychosocial Positive Resource Factors (PPRFs)

The predominant response trend was concentrated in the options frequently and very frequently, suggesting that the response density is oriented toward the right of the scaling, which is somewhat congruent with the negatively skewed distributions that were detected. This form of skewness occurred in all items (except for the recoding items), and kurtosis was found to be more variable across items; overall, statistical normality was not met by any of the items.

### 3.2. Use of Response Categories

#### 3.2.1. Psychosocial Risk Factors (PSRFs)

It was observed that the number of categories in the 22 PSRF items with fewer than 10 options (O < 10) and the AENO correlated linearly at −0.928, suggesting high convergence between the two metrics. Except for four PSRF items (18.1%; items 4, 9, 12 and 22) with efficient responses between 1 and 7 options, the rest produced between one and five options with fewer than 10 responses. The high rate of unused options occurred in the items referring to social interaction between workers (i.e., harassment at work and stressful leadership) and the activity schedule (working hours, shift and schedule).

#### 3.2.2. Psychosocial Positive Resource Factors (PPRFs)

Efficient AENO response options ranged from 3.8 (item 27) to 5.99 (item 24), predominantly with 5 efficient options (M_AENO_ = 4.99). Approximately half of the items (11, 57.8%) produced only one inefficient response option, while the rest produced between two and four frequencies below 10 responses. Compared to the PSRF section, this section contained fewer inefficient options. The concordance between O < 10 and the AENO among the 19 PPRF items (favorable resources) was *r* = −0.883 (*p* < 0.05), which is similar to what was found in the previous section (PSRFs). This association can be considered strong.

### 3.3. Distribution of Scaling

#### 3.3.1. Psychosocial Risk Factors (PSRFs)

The results are presented in Table 2. The AJUS system identified the following proportion of distributional forms: A = 6 (27.2%), J = 4 (18.1%), U = 10 (45.4), S = 2 (9%) and L = 0. Items identified with the distributional form S (multimodal) were predominant, indicating possible different response processes in these items. However, a discrepancy was detected between this type of distribution and its high response concentration (measured by the *TW* coefficient). To verify this apparent multimodality, correspondence was seen between the observation of distributional density plots (not reported here due to space limitations) and the Tw coefficients, and a clear regularity of item response distribution was found.

#### 3.3.2. Psychosocial Positive Resource Factors (PPRFs)

The results are shown in Table 3. Distributional forms were identified with the following frequency: A = 6 (31.5%); J = 1 (5.2%); U = 0; S = 11 (57.8%) and L = 1 (5.2%). These results indicate that approximately half had an apparent irregular distribution. As in the PSRF section above, the predominance of multimodal distributions (S) disagreed with the response concentration measured by *TW*: For these items, *TW* ranged from 0.55 to 0.72, indicating a moderate-to-high response concentration. However, observation of the density plots of these items (not shown here due to space limitations) with apparent distributional irregularity showed that the distributions are unimodal.

### 3.4. Association with External Variables

#### 3.4.1. Psychosocial Risk Factors (PSRFs)

Results are presented in Table 4. Gender and age maintained associations around zero or below 0.18, and all were statistically nonsignificant. With respect to the expected relationships with other constructs, except for the cognitive-attentional demand items (prolonged attention to tasks, two or more tasks and high mental effort) and work rhythm demands (amount and speed), the rest of the contents showed theoretical consistency, that is, negative associations with self-efficacy (OSES; *r_median_* = −0.22) and engagement (UWES; *r_median_* = −0.22), and positive associations with mental health disorders (PHQ-4; *r_median_* = 0.16). With the single item of stress, the direction was theoretically and consistently positive (*r_median_* = 0.10). In the work rhythm content area, item 3 (interruptions to complete tasks on time) was inconsistent with respect to the other two items in its content area; similarly, workday, shift and schedule item 6 (overtime) showed inconsistency with respect to the other two items in its content area.

#### 3.4.2. Psychosocial Positive Resource Factors (PPRFs)

Results are presented in Table 5. Gender and age maintained associations around zero and were statistically nonsignificant. The median magnitude of the monotonic associations remained in the following order: UWES (*r_median_* = 0.56), OSES (*r_median_* = 0.43), PHQ-4 (*r_median_* = −0.35) and SUI (*r_median_* = −0.03). Overall, (a) the direction of the association was theoretically consistent across all items with each external variable, and (b) the associations with all external variables were found to not vary substantially: In the UWES, OSES and PHQ, the high predominant size can be considered large, taking into account the categorical nature of the variables. With the stress measure, the monotonic association was trivial.

## 4. Discussion

In the framework of one of the stages of content validity assessment at the item level, in the present study, evidence of the possible application of the PROPSIT [43,44] to the Peruvian context and, given the etic approach of its constructs, potentially to other contexts was presented for the first time. This measure was originally created for the Mexican context and represents an advance in the integration of PWF models with more international evidence (effort–reward imbalance, [74]; demand–control, [75]; demands–resources, [47]) and cultural appropriateness coming from PWFs in Mexico [43] but with potential cross-cultural generalization. In general, the present study reveals evidence of a set of properties that are usually not reported in the validation of a measure applied to the measurement of PWFs. These properties focused on items with respect to a part of their structural characteristics and the association with external constructs.

Regarding the response trend in the items, a clear pattern was observed where PSRFs were less intense than favorable resource factors, indicating the predominance of one set of factors over others and the ability of the items to differentiate them. This overall differentiation also suggests that even with the variability in jobs and the work contexts sampled, the tendency is to find environments where favorable resources are perceived as more frequent than PSRFs.

In relation to scaling, the responses were not normally distributed, and this distortion came from the third and fourth moments of the distribution (i.e., skewness and kurtosis). The degree of distortion indicated by these moments does not appear to be strong and does not go beyond the limits of kurtosis and skewness [76]. This nonnormality seems to be the rule rather than the exception of social science measurements [77,78] and cannot represent a problem for characterizing the measured behaviors or for the parametric modeling of the whole instrument. Based on how the data will be treated in the analysis, structural equation modeling (SEM) parameterizations for items treated as categorical variables are effective with estimators such as diagonally weighted least squares (DWLS) or unweighted least squares (ULS) [79] and when treated as continuous variables [80], and they are currently reasonable estimators of item parameters. One implication is that the joint treatment of the PROPSIT items in their expected dimensions requires estimators that take into account this predictable nonnormality; additionally, considering the extent of scaling (i.e., seven response options), assumptions of response continuity can be considered [76,80]. This scaling should be further evaluated, because our results showed a possible modification, consisting of reducing the range to 4 or 5 options (derived from the actual equivalent number of options, AENO). Accordingly, and due to the prevalence of the number of efficient response options, further exploration should be performed.

With respect to the association results, the substantiality of the results shows, in general, a theoretical concordance of the contents with the external variables that were chosen. In this sense, the pattern of positive and negative monotonic associations detected in both areas (PSRFs and favorable psychosocial resources) indicated that the indicators hold theoretical and practical significance. Although the predominant magnitude of the associations obtained may be considered small to moderate compared to suggested interpretative guidelines (e.g., [81]), these guidelines were constructed to be applied to continuous variables and cannot be directly translated to our results to either qualify or estimate statistical power [82]. The interpretation of item analyses for the purpose of validating a measure requires adaptation of such guidelines, and applied to our results, the negative or positive direction of the correlations can be prioritized until a clear scheme for interpreting the magnitudes obtained is developed. However, even with this limitation, the size of the correlations obtained is an indicator of satisfactory item validity.

One methodological issue involved in the analysis of the above association requires attention. A general measure of dependence based on information theory (MIC coefficient) was applied in the study to detect some functional form between the variables, but it predominantly yielded results without statistical significance, i.e., the association did not present any functional form. On the one hand, the MIC coefficients of general association did not seem to have statistical power to detect any functional form of dependence between variables. On the other hand, monotonic associations between variables were detected by the Spearman coefficient of monotonicity, suggesting that ordinal categorization can create strong entropy that prevents us from distinguishing dense (i.e., “noise”) and uniform bivariate patterns. Specifically, regarding the substantive results derived from the monotonic correlation, risk factors and favorable resource items covaried more strongly with measures of engagement and distress (UWES-3 and PHQ-4, respectively) than the associations obtained with stress (SUI).

The items that showed less associative weakness or those that showed counterintuitive associations (cognitive-attentional demands such as prolonged attention to tasks, to two or more tasks and high mental effort as well as workload and pace demands such as the amount and speed of work tasks) require further exploration to discern the factors that influenced this type of result. We conjecture that the intensity of work demands assessed in the sample was restricted to low intensities, with high demands rather than episodic demands [38], such that they produced a manageable experience to solve them. This can be corroborated in the response trend, where options indicating lower intensities of these stressors showed a higher response density. However, other unassessed aspects that may have moderated this, such as social desirability, may have been partially responsible, and they require future exploration regarding their likely replicability and explanation.

A general implication of the present study is that in the practice of publications on the validity of psychosocial factor measures, there is an apparent need for space for detailed analyses at the item level to support complementary conclusions on the psychometric characterization of the measures studied, obtained by determining the factorial structure and its reliability. For reasons of space and desired emphasis in the exploration, such item-level analysis is not always convenient to present within the same publications.

Some suggestions for improving the PROPSIT in the Peruvian sample can be implemented but are conditional on the replicability of the results of the present study. First, these changes can focus on reducing the response scaling to five or six points, a modification that is not so distant from the original seven-point scaling. Incorporating these aspects will achieve a more comprehensive balance of the psychometric properties of various measures. Because item scaling was not performed in the study with the Mexican sample [43], it is uncertain whether this recommendation is relevant for other contexts. Second, items that produced theoretically different associations than expected should be re-evaluated to decide whether they should be replaced by equivalent content or reformulate their conceptualization as a psychosocial risk factor in Peruvian workers.

The present study has some limitations. First, the sample size could be a conditioning factor for the distribution of responses in the options and will increase the type I error in the identification of unusable categories. Thus, a sample size with sufficient statistical power to overcome the minimum frequency of 10 responses is highly recommended. Second, the representativeness of the sample with respect to the sampled occupations is not guaranteed because the study design was oriented toward maximizing the heterogeneity of the occupations. Third, relatedly, this heterogeneity could also be a determinant of the emergence of underutilized response options, for example, when some PSRFs are strongly linked to the intensity of such response options, such as physical effort. Fourth, other aspects of the quality of the PROPSIT ordinal system were not evaluated. They correspond to a set of quality indicators [40], such as the distance between thresholds, the monotonicity of the item–score relationship and the monotonic progression of the response options. However, this type of evaluation requires defining the dimensionality of each item as a fundamental assumption and fitting a parametric model, such as the partial credit model. At this stage of the research, the focus was on the properties of the items as individual items, linked by theoretical constructs. Inspecting dimensionality in a rigorous manner requires extending the length of this manuscript by 50% or more, and to avoid false positives in identifying dimensionality, this goal remains for future research. Fifth, measures based on single items have advantages and limitations, and in relation to the SUI [52], it is required to discern between the lack of validity of this item for the Peruvian context in general, the absence of statistical power and its lack of sensitivity to the contents sampled in the PROPSIT. Finally, social desirability was not included as another external variable, and including it in the design is highly recommended to evaluate the possible impact of this response style.

## 5. Conclusions

Since measures describing the psychosocial environment serve to make decisions about its functioning in relation to its effects on workers, the content of these measures needs to be carefully and extensively analyzed to assess their quality. The study essentially reached some conclusions. First, the relationships between the items of the PROPSIT and external measures maintain theoretically consistent and low-to-moderate relationships for the content defined as favorable job resources, while for psychosocial risk factors, there were some discrepancies that may stem from sample idiosyncrasies or social desirability biases. Third, it is clear that research on the construct validity of the items should incorporate not only the strength of the item–construct relationship but also the association with external constructs or behavioral criteria, which have consequences for the content validity and the interpretation of the scores to be obtained.

## Figures and Tables

**Table 1 ijerph-19-07972-t001:** Descriptive statistics of the effective study sample.

Variable	N	%
Gender		
Male	96	51.1
Female	92	48.9
Marital Status		
Married	37	19.7
Cohabitant	24	12.8
Divorced	3	1.6
Single	124	124
Contract		
Planta/Definitive	95	50.5
Eventual/Temporal	93	49.5
Job level		
Director, manager, supervisor or area manager	27	14.4
Administrative, nonmanagerial (nonmanual work)	101	53.7
Sales, without personnel in charge (nonmanual work)	9	4.8
Operative employee (manual work)	49	26.1
Other	2	1.1
Convivencia		
Alone	11	5.9
Alone with a pet	1	0.5
Partner or family	171	91.0
With friends	3	1.6
Others	2	1.1
Classification of academic areas		
Health sciences	6	3.2
Basic sciences	1	0.5
Engineering	20	10.6
Economic and management sciences	65	34.6
Humanities, law and social sciences	39	20.7
Does not belong/not applicable	56	29.8
Missing	1	0.5
Occupational classification CIUO-08		
Directors and managers	3	1.6
Scientific and intellectual professionals	58	30.9
Technicians and mid-level professionals	57	30.3
Administrative support personnel	30	16.0
Service workers, store and market salespersons	8	4.3
Military, craftsmen, mechanical and other trades workers	26	13.8
Plant and machine operators and assemblers	4	2.1
Elementary occupations	1	0.5

**Table 2 ijerph-19-07972-t002:** Descriptive statistical results of the psychosocial risk factor items.

Psychosocial Risk Factors	M	SD	Sk	Ku	AD	AJUS	TW	O < 10	AENO
Workload and work rhythm demands									
1. Workload	4.67	1.53	−0.57	−0.85	9.96	A	0.62	1, 7	5.29
2. Work fast	4.74	1.56	−0.64	−0.57	8.24	S	0.62	1	5.8
3. Interruptions for on-time tasks	2.83	1.44	0.94	0.38	9.05	A	0.66	7	5.17
High responsibility demands									
4. Take important actions	2.54	1.72	1.00	−0.09	13.06	L	0.56	0	5.20
5. Hazards	2.07	1.32	1.53	2.22	15.60	U	0.71	5, 6, 7	4.07
Shift and schedule demands									
6. Overtime, long hours	3.30	1.49	0.28	−0.88	5.36	S	0.64	7	5.71
7. Rotation/change of shift	1.72	1.31	1.93	2.94	33.09	S	0.71	4, 6, 7	2.99
8. Working at night	1.58	1.24	2.45	5.81	4.22	U	0.73	4, 5, 6, 7	2.55
Cognitive/attentional demands									
9. Prolonged attention to tasks	4.53	1.74	−0.55	−0.92	8.76	S	0.55	0	5.95
10. Attending to two or more tasks	4.64	1.66	−0.59	−0.79	9.04	A	0.58	1	5.75
11. Mental effort	4.60	1.65	−0.37	−0.96	6.00	A	0.58	1	6.09
Emotional demands									
12. Negative emotions of other people	3.51	1.58	0.62	−0.39	6.39	A	0.62	0	5.88
13. Dealing with people	2.51	1.25	1.14	1.47	1.27	S	0.72	6, 7	4.45
14. Show different emotions	2.64	1.55	1.15	0.65	11.73	A	0.63	6, 7	5.09
Physical effort demands									
15. A lot of physical effort	2.53	1.66	1.01	−0.10	13.39	S	0.59	7	5.04
16. Uncomfortable positions	2.28	1.53	1.18	0.19	17.30	S	0.54	7	4.35
17. Adverse environmental conditions	2.17	1.34	1.21	0.83	13.79	L	0.70	6, 7	4.34
Harassment at work									
18. Psychological abuse (boss, supervisors)	1.26	0.76	4.39	22.30	47.61	U	0.80	3, 4, 5, 6, 7	1.80
19. Psychological abuse (peers)	1.21	0.73	5.14	29.29	53.60	U	0.81	3, 4, 5, 6, 7	1.59
Stressful leadership									
20. Too much control	1.84	1.02	2.08	5.95	17.80	S	0.80	4, 5, 6, 7	3.18
21. Exaggerated rules and regulations	1.96	1.14	1.73	3.22	16.77	S	0.72	4, 5, 6, 7	3.52
22. Adequate feedback from supervisors (R)	4.69	1.69	0.75	−0.48	9.59	S	0.58	0	5.71

Note. Sk and Ku: Skewness and kurtosis coefficients. AD: Anderson–Darling normality test; AJUS: Distribution classification; TW: Concentration coefficient; AENO: Actual equivalent number of options. O < 10: Response options with a frequency of less than 10.

**Table 3 ijerph-19-07972-t003:** Descriptive statistical results of the psychosocial positive resource items.

Psychosocial Positive Resources	M	SD	Sk	Ku	AD	AJUS	TW	O < 10	AENO
Rewards and career development									
1. Fair and equitable work	4.85	1.58	−0.72	−0.41	9.19	S	0.61	7	5.48
2. Motivating salary	4.74	1.76	−0.63	−0.77	8.91	S	0.55	0	5.99
3. Work valued and recognized	4.92	1.46	−0.75	−0.26	9.08	S	0.66	1	5.24
4. No professional growth opportunities (R)	2.27	1.33	1.05	0.58	11.25	L	0.69	5, 6, 7	4.50
5. Loss of employment	2.04	1.26	1.66	2.70	16.12	S	0.73	4, 5, 6, 7	3.88
6. Pleasant and rewarding work	5.20	1.43	−0.89	0.11	9.67	A	0.66	1	5.09
7. Tasks benefit people and society	5.28	1.40	−0.90	0.20	9.18	A	0.67	1, 2	5.07
Labor control and task content									
8. Free to decide the job	5.18	1.48	−0.94	0.13	10.23	A	0.65	1	5.16
9. I use skills	5.68	1.38	−1.17	0.64	13.32	S	0.68	1, 2	4.49
10. Capacity building	5.67	1.42	−1.08	0.32	12.92	J	0.66	1, 2	4.95
11. Very varied activities	5.41	1.49	−0.85	−0.32	11.40	S	0.63	1, 2	4.60
12. Very clear roles and tasks	5.62	1.39	−1.16	0.75	12.59	S	0.67	1, 2	4.64
Resources to carry out the work									
13. Necessary and appropriate materials	5.58	1.49	−1.28	0.95	14.34	S	0.65	1	5.49
14. Necessary training	5.01	1.64	−0.91	−0.10	10.46	S	0.60	1	4.99
Workplace climate and social support									
15. Partner support	5.10	1.36	−0.80	0.32	8.23	A	0.70	1	5.11
16. Supervisor support	5.05	1.34	−0.57	−0.08	6.68	A	0.70	1	4.93
17. Climate of union/collaboration	5.22	1.34	−0.79	0.18	8.19	S	0.69	1	4.87
Value congruence									
18. Values match with the organization	5.09	1.36	−0.80	0.03	9.11	S	0.62	1	5.17
19. Values match with peers	4.96	1.41	−0.87	0.20	8.96	A	0.68	1	5.17

Note. Sk and Ku: Skewness and kurtosis coefficients. AD: Anderson–Darling normality test; AJUS: Distribution classification; TW: Concentration coefficient; AENO: Actual equivalent number of options. O < 10: Response options with a frequency of less than 10.

**Table 4 ijerph-19-07972-t004:** Associative results of the psychosocial risk factor items.

**Factores de Riesgo Psicosocial (RP)**	SIS (Stress)		OSES (Self-Efficacy)		UWES (Engagement)		PHQ-4 (Distress)		Gender	Age
	MIC	*rho*	MIC	*rho*	MIC	*rho*	MIC	*rho*		
Workload and work rhythm demands										
1. Workload	0.083	** *0.140* **	0.136	0.222	0.159	0.343	0.204	−0.289	0.046	0.069
2. Work fast	0.101	** *0.175* **	0.142	** *0.144* **	0.119	0.261	0.173	−0.355	0.076	0.006
3. Interruptions for on-time tasks	0.061	0.047	0.190	−0.305	0.155	−0.407	0.245	0.486	−0.104	−0.006
High responsibility demands										
4. Take important actions	0.045	0.00	0.104	** *−0.176* **	0.103	−0.225	0.056	0.23	0.112	0.037
5. Hazards	0.038	−0.027	0.152	−0.292	0.138	−0.303	0.082	** *0.175* **	** *0.171* **	0.015
Shift and schedule demands										
6. Overtime, long hours	0.05	0.097	0.116	0.007	0.107	** *0.140* **	0.071	** *−0.178* **	−0.029	−0.072
7. Rotation/change of shift	0.043	0.067	0.146	−0.303	0.164	−0.290	0.084	** *0.142* **	0.00	−0.002
8. Working at night	0.052	0.024	0.160	−0.343	0.156	−0.353	0.107	0.283	0.074	0.058
Cognitive/attentional demands										
9. Prolonged attention to tasks	0.086	0.096	0.169	0.285	0.152	0.319	0.206	−0.420	0.085	0.110
10. Attending to two or more tasks	0.082	** *0.140* **	0.134	0.243	0.146	0.290	0.191	−0.361	0.076	0.011
11. Mental effort	0.041	0.100	0.228	0.416	0.181	0.439	0.122	−0.295	0.019	0.089
Emotional demands										
12. Negative emotions of other people	0.061	0.046	0.129	−0.042	0.112	−0.117	0.099	0.242	** *−0.176* **	0.028
13. Dealing with people	0.039	** *0.175* **	0.139	−0.254	0.110	−0.249	0.09	0.214	−0.110	0.016
14. Show different emotions	0.064	** *0.176* **	0.150	−0.324	0.150	−0.353	0.232	0.432	−0.145	−0.006
Physical effort demands										
15. A lot of physical effort	0.052	** *0.164* **	0.143	−0.188	0.112	−0.217	0.064	0.135	0.103	0.018
16. Uncomfortable positions	0.061	0.117	0.133	−0.212	0.119	−0.212	0.059	0.129	0.082	−0.047
17. Adverse environmental conditions	0.070	0.120	0.187	−0.229	0.107	−0.192	0.129	0.204	** *0.168* **	0.005
Harassment at work										
18. Psychological abuse (superiors)	0.028	** *0.137* **	0.184	−0.347	0.117	−0.307	0.107	0.246	0.005	0.039
19. Psychological abuse (peers)	0.024	0.036	0.114	−0.256	0.117	−0.272	0.089	0.216	0.005	0.118
Stressful leadership										
20. Too much control	0.050	** *0.144* **	0.143	−0.274	0.106	−0.253	0.129	*0.151*	−0.012	0.121
21. Exaggerated rules and regulations	0.044	0.115	0.124	−0.233	0.151	−0.288	0.112	** *0.180* **	** *0.154* **	0.056
22. Adequate feedback from supervisors (R)	0.078	0.080	0.194	0.345	0.290	0.549	0.234	−0.397	−0.007	** *−0.202* **

Note. SIS: Single-item stress; OSES: Occupational self-efficacy; UWES: Work engagement; PHQ-4: Psychological distress; MIC: Maximum information coefficient; *rho*: Spearman monotonic correlation. In bold and italics: *p* < 0.05. Underlined: *p* < 0.002 (Bonferroni correction in *α* = 0.05). R: Reversed content item.

**Table 5 ijerph-19-07972-t005:** Associative results of the psychosocial positive resource items.

Positive Psychosocial Resources	SIS (Stress)		OSES (Self-Efficacy)		UWES (Engagement)		PHQ-4 (Distress)		Gender	Age
	MIC	*rho*	MIC	*rho*	MIC	*rho*	MIC	*rho*		
Rewards and career development										
1. Fair and equitable work	0.057	0.00	0.269	0.443	0.267	0.561	0.212	−0.394	0.090	−0.060
2. Motivating salary	0.080	−0.030	0.222	0.437	0.298	0.649	0.308	−0.579	0.110	−0.040
3. Work valued and recognized	0.060	0.017	0.264	0.447	0.291	0.612	0.201	−0.358	0.060	−0.100
4. No professional growth opportunities (R)	0.068	0.00	0.161	−0.333	0.204	−0.447	0.158	0.419	−0.070	0.140
5. Loss of employment	0.032	−0.038	0.171	−0.326	0.171	−0.408	0.138	0.277	0.060	−0.060
6. Pleasant and rewarding work	0.046	−0.048	0.239	0.457	0.394	0.674	0.21	−0.378	0.020	0.000
7. Tasks benefit people and society	0.040	0.041	0.141	0.353	0.247	0.433	0.086	** *−0.186* **	−0.070	−0.030
Labor control and task content										
8. Free to decide the job	0.046	−0.057	0.167	0.411	0.298	0.513	0.136	−0.26	0.000	0.060
9. I use skills	0.027	0.000	0.184	0.477	0.389	0.634	0.189	−0.371	0.000	−0.040
10. Capacity building	0.028	0.013	0.231	0.511	0.385	0.684	0.183	−0.376	0.010	−0.020
11. Very varied activities	0.020	0.027	0.219	0.513	0.321	0.652	0.165	−0.395	0.010	0.020
12. Very clear roles and tasks	0.047	−0.010	0.185	0.458	0.340	0.632	0.209	−0.393	−0.050	0.030
Resources to carry out the work										
13. Necessary and appropriate materials	0.082	−0.110	0.169	0.435	0.345	0.582	0.241	−0.421	0.080	0.000
14. Necessary training	0.086	−0.067	0.253	0.454	0.298	0.591	0.239	−0.407	0.040	−0.070
Workplace climate and social support										
15. Partner support	0.066	−0.090	0.195	0.310	0.293	0.453	0.107	** *−0.177* **	0.050	−0.120
16. Supervisor support	0.057	−0.052	0.188	0.354	0.274	0.478	0.132	** *−0.159* **	0.03	−0.100
17. Climate of union/collaboration	0.049	−0.073	0.227	0.365	0.303	0.52	0.141	−0.259	0.130	−0.030
Value congruence										
18. Values match with the organization	0.047	−0.051	0.220	0.385	0.344	0.544	0.161	−0.304	0.090	−0.070
19. Values match with peers	0.053	0.036	0.170	0.348	0.253	0.497	0.170	−0.289	0.030	−0.070

Note. SIS: Single-item stress; OSES: Occupational self-efficacy; UWES: Work engagement; PHQ-4: Psychological distress; MIC: Maximum information coefficient; *rho*: Spearman monotonic correlation. In bold and italics: *p* < 0.05. Underlined: *p* < 0.002 (Bonferroni correction in *α* = 0.05).

## Data Availability

The raw data supporting the conclusions of this article will be made available by the authors, without undue reservation.

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
