# Peer review of "Item-Level Psychometric Analysis of the Psychosocial Processes at Work Scale (PROPSIT) in Workers"

_ijerph, 2022, doi:10.3390/ijerph19137972_

Round 1

Reviewer 1 Report

General comment

This study aimed to evaluate the item properties of a measure of psychosocial work factors(PWFs). To reach this goal, the paper collected data through a web platform from 188 Peruvian working adults (men = 101, 50.5%) using the Psychosocial Processes at Work Scale (PROPSIT) and analyzed the distributional characteristics, the efficiency of its response options and its correlates with engagement, occupational self-efficacy, general stress and psychological distress.

The paper indicates that the items are asymmetrically distributed, without statistical normality and with a response tendency at low (for psychosocial risk factors [PSRFs]) and medium (favorable psychosocial resources) levels and discusses the contributions of the results to the content validity of the PROPSIT and the orientation of working hypotheses about PROPSIT item constructs and measures of work effects. However, there are still some problems.

Specific comments

Format and copyright

  1. It’s difficult to distinguish the italics in table4 and table 5. It is suggested to change to another representation. (p13-15)

Content

  1. The full name of FPST should be introduced. (p5, line 185)
  2. In discussion, more suggestions on how to improve the scale can be put forward.
  3. It is suggested to write 6-8 sentences in one paragraph. Some paragraphs are too short or too long. (p17, line 489-512)
  4. The conclusion is too long, lacking generalization. The second and third points of the conclusion are suggestions and outlooks, which can be put into the discussion. (p17-18, line 514-528)

Author Response

Review Report (Reviewer 1)

English language and style

( ) Extensive editing of English language and style required
( ) Moderate English changes required
( X) English language and style are fine/minor spell check required
( ) I don't feel qualified to judge about the English language and style

Comments and Suggestions for Authors

General comment

This study aimed to evaluate the item properties of a measure of psychosocial work factors(PWFs). To reach this goal, the paper collected data through a web platform from 188 Peruvian working adults (men = 101, 50.5%) using the Psychosocial Processes at Work Scale (PROPSIT) and analyzed the distributional characteristics, the efficiency of its response options and its correlates with engagement, occupational self-efficacy, general stress and psychological distress.

The paper indicates that the items are asymmetrically distributed, without statistical normality and with a response tendency at low (for psychosocial risk factors [PSRFs]) and medium (favorable psychosocial resources) levels and discusses the contributions of the results to the content validity of the PROPSIT and the orientation of working hypotheses about PROPSIT item constructs and measures of work effects. However, there are still some problems.

Specific comments

It’s difficult to distinguish the italics in table4 and table 5. It is suggested to change to another representation. (p13-15)

Response:

Thank you for pointing out your observation.

Change:

Each number in italics was modified, now in bold and italics. A footnote was also added: “In bold and italics… “

  1. The full name of FPST should be introduced. (p5, line 185)

Response:

Thank you for this observation. We made the change.

Change:

We eliminated FPST, and replaced it with “psychosocial factors at work”.

  1. In discussion, more suggestions on how to improve the scale can be put forward.

Response:

Thank you for this observation. We made the change.

Change:

Some suggestions for improving the PROPSIT in the Peruvian sample can be implemented, but conditional on the replicability of the results of the present study. First, these changes can focus on reducing the response scaling to five or six points, a modification that is not so distant from the original seven-point scaling. Incorporating these aspects will achieve a more comprehensive balance of the psychometric properties of various measures. Because item scaling was not studied in the study with the Mexican sample [44], it is uncertain whether this recommendation is relevant for other contexts. Second, items that produced theoretically different associations than expected should be re-evaluated to decide whether they should be replaced by equivalent content or reformulate their conceptualization as a psychosocial risk factor in Peruvian workers.

  1. It is suggested to write 6-8 sentences in one paragraph. Some paragraphs are too short or too long. (p17, line 489-512)

Response:

Thank you for pointing out your observation. However, each sentence is brief or long in relation to whether we detail sufficiently what we mean. Since our research is technical, we have tried to be very explicit in our ideas. If we separate each sentence, we believe that what we want to communicate is clear and condensed. Here are the separate sentences, but we made a small reduction (see Changes):

“Thus, a sample size with sufficient statistical power to overcome the minimum frequency of 10 responses is highly recommended.

Second, the representativeness of the sample with respect to the sampled occupations is not guaranteed because the study design was oriented toward maximizing the heterogeneity of the occupations.

Third, relatedly, this heterogeneity could also be a determinant of the emergence of underutilized response options, for example, when some PSRFs are strongly linked to the intensity of such response options, such as physical effort.

Fourth, other aspects of the quality of the PROPSIT ordinal system were not evaluated, like a set of quality indicators [41]such as the distance between thresholds, the monotonicity of the item-score relationship, and the monotonic progression of the response options.

However, this type of evaluation requires defining the dimensionality of each item as a fundamental assumption and fitting a parametric model, such as the partial credit model.

At this stage of the research, the focus was on the properties of the items as individual items, linked by theoretical constructs.

Inspecting dimensionality in a rigorous manner requires extending the length this manuscript by 50% or more, and to avoid false positives in identifying dimensionality, this goal remains for future research.

Fifth, measures based on single items have advantages and limitations, and in relation to the SUI [53], it is required to discern between the lack of validity of this item for the Peruvian context in general, the absence of statistical power, and its lack of sensitivity to the contents sampled in the PROPSIT.

Finally, social desirability was not included as another external variable, and including it in the design is highly recommended to evaluate the possible impact of this response style.”

Change:

We replace this:

Fourth, other aspects of the quality of the PROPSIT ordinal system were not evaluated. They correspond to a set of quality indicators [41],

… with this.

“Fourth, other aspects of the quality of the PROPSIT ordinal system were not evaluated, like a set of quality indicators [41], …”

  1. The conclusion is too long, lacking generalization. The second and third points of the conclusion are suggestions and outlooks, which can be put into the discussion. (p17-18, line 514-528)

Response:

Thank you for pointing out your observation. We moved the content referring to suggestions to the discussion, and improved the conclusions.

Change:

The study essentially reaches some conclusions: First, the relationships between the items of the PROPSIT and external measures maintain theoretically consistent and low-to-moderate relationships for the content defined as favorable job resources, while for psychosocial risk factors, there were some discrepancies that may stem from sample idiosyncrasies or social desirability biases. And third, it is clear that research on the construct validity of the items should incorporate not only the strength of the item-construct relationship but also the association with external constructs or behavioral criteria, which have consequences for the content validity and the interpretation of the scores to be obtained.

Reviewer 2 Report

The manuscript entitled "Item-Level Psychometric Analysis of the Psychosocial Processes at Work Scale (PROPSIT) in Workers" is an interesting validation study, showing an adaptation of PROPORIT to Peruvian culture. The main question of the research was to assess the item properties of the PROPSIT, regards their structural attributes and correlates. Most validation studies are only interested on the stucture of measure (by using EFA, CFA) or composite scores properties of the scales and subscales of a questionnaire  (e.g., associations between items,  discrimination, relibility, by using CR, AVE, or HTMT statistics). This study is quite original, since the authors choosed a modern and relatively rare item-level approach, which can be consider a current alternative to the classical factor analysis or item response theory. Items were treated as ordinal categorical variables to examine their distributional properties, such as response trend, efficiency of use response categories (assessed by actual equivalent number of options, AENO), and scaling distribution pattern (evaluated by the AJUS). Construct validity was evaluated using correlation analysis with several external variables, including demographics (gender, age), occupational self-efficacy, work engagement, stress, and depresion.

The topic is important and relevant in the field, because there is a contonuous need to measure well-being in work by using a good and reliable  tools. The Psychosocial Processes at Work Scale was developed to assess psychosocial work conditions and the effects on the well-being of the workers, regards factors, effects and intervening processes of work. The PROPSIT consists of thwo scales to measure psychosocial risk factors and psychosocial positive resources, and seems a good alternative to the other work-related questionnaires.

The title is adequate. The Abstract clearly describes all parts of the manuscript. The Introduction section is comprehensive and concise. Material and methods are well-written in all parts, so replication of this study is possible. Statistics are adequately selected and accurate. Results are clearly presented in tables, consistent with the highest scientific standards. Tables are well-arranged, clear, transparent and informative. Discussion is interesting, limitation of the study exhaustive and complete. The conclusions are consistent with the evidence and arguments presented and they address the main question posed. The references are appropriate to the content. I can't find anything that can be improved.

Author Response

Review Report (Reviewer 2)

English language and style

( ) Extensive editing of English language and style required
( ) Moderate English changes required
() English language and style are fine/minor spell check required
( X) I don't feel qualified to judge about the English language and style

Yes

Can be improved

Must be improved

Not applicable

Does the introduction provide sufficient background and include all relevant references?

(x )

( )

( )

( )

Is the research design appropriate?

( x)

( )

( )

( )

Are the methods adequately described?

( x)

( )

( )

( )

Are the results clearly presented?

( x)

( )

( )

( )

Are the conclusions supported by the results?

( X)

( )

( )

( )

Comments and Suggestions for Authors

The manuscript entitled "Item-Level Psychometric Analysis of the Psychosocial Processes at Work Scale (PROPSIT) in Workers" is an interesting validation study, showing an adaptation of PROPORIT to Peruvian culture. The main question of the research was to assess the item properties of the PROPSIT, regards their structural attributes and correlates. Most validation studies are only interested on the stucture of measure (by using EFA, CFA) or composite scores properties of the scales and subscales of a questionnaire  (e.g., associations between items,  discrimination, relibility, by using CR, AVE, or HTMT statistics). This study is quite original, since the authors choosed a modern and relatively rare item-level approach, which can be consider a current alternative to the classical factor analysis or item response theory. Items were treated as ordinal categorical variables to examine their distributional properties, such as response trend, efficiency of use response categories (assessed by actual equivalent number of options, AENO), and scaling distribution pattern (evaluated by the AJUS). Construct validity was evaluated using correlation analysis with several external variables, including demographics (gender, age), occupational self-efficacy, work engagement, stress, and depresion.

The topic is important and relevant in the field, because there is a contonuous need to measure well-being in work by using a good and reliable  tools. The Psychosocial Processes at Work Scale was developed to assess psychosocial work conditions and the effects on the well-being of the workers, regards factors, effects and intervening processes of work. The PROPSIT consists of thwo scales to measure psychosocial risk factors and psychosocial positive resources, and seems a good alternative to the other work-related questionnaires.

The title is adequate. The Abstract clearly describes all parts of the manuscript. The Introduction section is comprehensive and concise. Material and methods are well-written in all parts, so replication of this study is possible. Statistics are adequately selected and accurate. Results are clearly presented in tables, consistent with the highest scientific standards. Tables are well-arranged, clear, transparent and informative. Discussion is interesting, limitation of the study exhaustive and complete. The conclusions are consistent with the evidence and arguments presented and they address the main question posed. The references are appropriate to the content. I can't find anything that can be improved.

Response:

We greatly appreciate your assessment of the manuscript, and your comments help us to maintain a path of creativity and thoroughness.

Change:

None

Reviewer 3 Report

Please, revise the first sentence "At the stage in which measures are created and/or adapted, the breakdown of the univariate information on the individual variables that constitute the composite scores is content that is considered very important not only for better understanding the quantitative behavior of scales or tests but also for complementary purposes such as replicability, meta-analytic evaluation, compliance with assumptions and, in general, good practices of reporting quantitative results"

It is too long with unclear phrases, such as  "quantitative behavior of scales or tests"

Revise the sentence "In this sense, the incorporation of modern methods for detecting a wide range of bivariate associations in the behavioral sciences also expansive opportunities for application when adapting of psycho-social measures of the work environment." 

It seems incomplete, unfinished.

First you state about conducting your study in Peruvian context, later - about Mexico. Precise where the psychometric properties of the new scale are established.

"In the present study, the methodological aspects above (i.e., the exploration of non-linear associations, the nomological network and the quality of the ordinal system) were applied to adapt a new measure of work factors and processes, the scale of the Psychosocial Processes at Work Scale (PROPSIT; [44,45]), to the Peruvian context."

"The PROPSIT 137 was created to represent the main factors of the work context (e.g., psychological demands, control, rewards) extralabor factors (e.g., family stressors, traffic transfer), mediating psychological effects (e.g., burnout, engagement) and relevant mental health variables (e.g., somatic symptoms and alterations including depression, anxiety, posttraumatic stress) in Mexican workers."

"The theoretical framework for this measure was the demands-resources model [46–48]. Additionally, the measure was constructed with a culturally relevant view to its Mexican context of origin, but it is potentially generalizable to other Latin American contexts due to the theoretical analysis of factors with etic value (i.e., factors consistently identified in several international studies; [49]) and that are consistently chosen as core attributes to assess PSRFs."

"Given the above, the aim of the present study was to identify the structural properties of the items of the PROPSIT work factors scale as well as their convergent and divergent associations with constructs of psychosocial effects on Peruvian workers."

Please, explain how the sample was selected and why only 188 workers were studied. 

Explain the abbreviation FPST in the sentence "The present study validated the FPST section, consisting of two major parts: favorable resources..."

because here you introduce it first.

Please, indicate how many items measure generalized anxiety in the sentence:

"The PHQ-4 is a brief screening measure of emotional and cognitive symptoms of depression (two items) and generalized anxiety (items), and it has been internationally accepted as a total screening measure of efficient psychological distress."

Revise the phrase "It is scaled with five items (from not at all to almost every day)..." It may be:

The answers are given on a five-point scale (from not at all to almost every day)

Why the research was approved in Mexico if it was conducted in Peru? "...was approved by the Research, Ethics, and Biosafety Commissions of the Hospital Infantil de México Federico Gómez, National Institute of Health, in Mexico City"

You state "The MVN program [65] was used." However, it is better to say:

The MVN statistical package from software R [65] was used.

You state "The agrmt program [70] was used for these analyses." It is better to say:

The agrmt statistical package from software R [70] was used for these analyses.

Author Response

Review Report (Reviewer 3)

English language and style

( ) Extensive editing of English language and style required
( ) Moderate English changes required
(x) English language and style are fine/minor spell check required
( ) I don't feel qualified to judge about the English language and style

Yes

Can be improved

Must be improved

Not applicable

Does the introduction provide sufficient background and include all relevant references?

( x)

( )

( )

( )

Is the research design appropriate?

( x)

( )

( )

( )

Are the methods adequately described?

( )

( X)

( )

( )

Are the results clearly presented?

( x)

( )

( )

( )

Are the conclusions supported by the results?

( X)

( )

( )

( )

Comments and Suggestions for Authors

Please, revise the first sentence "At the stage in which measures are created and/or adapted, the breakdown of the univariate information on the individual variables that constitute the composite scores is content that is considered very important not only for better understanding the quantitative behavior of scales or tests but also for complementary purposes such as replicability, meta-analytic evaluation, compliance with assumptions and, in general, good practices of reporting quantitative results"

It is too long with unclear phrases, such as  "quantitative behavior of scales or tests"

Response:

Thank you for pointing out your observation. We made the change.

Change:

In the creation and/or adaptation phase of the measures, detailed univariate item information is considered a good practice for reporting quantitative results, and serves to better understand the quantitative functioning of the scales or tests, and for the characterization of the measured construct.

Revise the sentence "In this sense, the incorporation of modern methods for detecting a wide range of bivariate associations in the behavioral sciences also expansive opportunities for application when adapting of psycho-social measures of the work environment." It seems incomplete, unfinished.

Response:

Thank you for pointing out your observation. We made the change.

Change:

In this sense, the incorporation of modern methods to detect a wide range of bivariate associations also expands the knowledge about the construct measured, as well as the performance of the items when adapting psychosocial measures of the work environment.

First you state about conducting your study in Peruvian context, later - about Mexico. Precise where the psychometric properties of the new scale are established.

"In the present study, the methodological aspects above (i.e., the exploration of non-linear associations, the nomological network and the quality of the ordinal system) were applied to adapt a new measure of work factors and processes, the scale of the Psychosocial Processes at Work Scale (PROPSIT; [44,45]), to the Peruvian context."

"The PROPSIT 137 was created to represent the main factors of the work context (e.g., psychological demands, control, rewards) extralabor factors (e.g., family stressors, traffic transfer), mediating psychological effects (e.g., burnout, engagement) and relevant mental health variables (e.g., somatic symptoms and alterations including depression, anxiety, posttraumatic stress) in Mexican workers."

"The theoretical framework for this measure was the demands-resources model [46–48]. Additionally, the measure was constructed with a culturally relevant view to its Mexican context of origin, but it is potentially generalizable to other Latin American contexts due to the theoretical analysis of factors with etic value (i.e., factors consistently identified in several international studies; [49]) and that are consistently chosen as core attributes to assess PSRFs."

"Given the above, the aim of the present study was to identify the structural properties of the items of the PROPSIT work factors scale as well as their convergent and divergent associations with constructs of psychosocial effects on Peruvian workers."

Response:

We regret that we seem confused in linking the development of PROPSIT with the new context of application (Peru). However, after a re-reading of the paper, we did not perceive any incongruence in our objective, because we are not stating that we are conducting the study in the Mexican context, but only in the Peruvian context. In this sense the psychometric properties were first established in Mexico, as we stated in the manuscript.

"The PROPSIT was created to represent the … in Mexican workers."

“Additionally, the measure was constructed with a culturally relevant view to its Mexican context of origin, but it is potentially …

Change:

None. 

Please, explain how the sample was selected and why only 188 workers were studied. 

Response:

  1. Thank you for this comment. For clarity, we added a conditional criterion to the application, referring to voluntary participation through participation consent form, and modified eligibility to selection.
  2. In the context of the time available for data collection, the voluntary nature of participation, and the efficiency of our procedure, the number of participants was the only one obtained. The limitations of the sample size and its representativeness are stated at the end of the manuscript.

Change:

The selection of the participants after data collection was based on a) contractual employment with a Peruvian employer in the last job, b) a minimum work time of six months, and c) voluntary acceptation for participate for mean participation consent form.

Explain the abbreviation FPST in the sentence "The present study validated the FPST section, consisting of two major parts: favorable resources..." because here you introduce it first.

Response:

Thank you for pointing out your observation. We made the change.

Change:

“The present study validated the psychosocial factors at work section, …”

Please, indicate how many items measure generalized anxiety in the sentence:

"The PHQ-4 is a brief screening measure of emotional and cognitive symptoms of depression (two items) and generalized anxiety (items), and it has been internationally accepted as a total screening measure of efficient psychological distress."

Response:

Thank you for pointing out your observation. We made the change.

Change:

“… generalized anxiety (two items), and …”

 Revise the phrase "It is scaled with five items (from not at all to almost every day)..." It may be: The answers are given on a five-point scale (from not at all to almost every day)

Response:

Thank you for pointing out your observation. We made the change.

Change:

 “The answers are given on a five-point scale …”

Why the research was approved in Mexico if it was conducted in Peru? "...was approved by the Research, Ethics, and Biosafety Commissions of the Hospital Infantil de México Federico Gómez, National Institute of Health, in Mexico City"

Response:

The main author's institution of affiliation is still building an ethics committee, and it is not available at the moment. This problem is characteristic of many Peruvian institutions of higher education.

Change:

None

You state "The MVN program [65] was used." However, it is better to say: The MVN statistical package from software R [65] was used.

Response:

Thank you for pointing out your observation. We made the change.

Change:

“The MVN statistical package from software R…”

You state "The agrmt program [70] was used for these analyses." It is better to say: The agrmt statistical package from software R [70] was used for these analyses.

Response:

Thank you for pointing out your observation. We made the change.

Change:

“The agrmt statistical package from software R …”

We greatly appreciate all the contributions of the reviewers that we are sure will enrich our manuscript.
